# Targeting Oral Squamous Cell Carcinoma with Combined Polo-Like-Kinase-1 Inhibitors and γ-Radiation Therapy

**DOI:** 10.3390/biomedicines12030503

**Published:** 2024-02-23

**Authors:** Subhanwita Sarkar, Ayan Chanda, Rutvij A. Khanolkar, Meghan Lambie, Laurie Ailles, Scott V. Bratman, Aru Narendran, Pinaki Bose

**Affiliations:** 1Department of Oncology, Cumming School of Medicine, University of Calgary, Calgary, AB T2N 4N1, Canada; subhanwita.sarkar@ucalgary.ca (S.S.); rutvij.khanolkar@ucalgary.ca (R.A.K.); a.narendran@ucalgary.ca (A.N.); 2Department of Biochemistry and Molecular Biology, University of Calgary, Calgary, AB T2N 4N1, Canada; achanda@ucalgary.ca; 3Princess Margaret Cancer Centre, University Health Network, Toronto, ON M5G 2C4, Canada; meglambie@gmail.com (M.L.); laurie.ailles@utoronto.ca (L.A.); scott.bratman@uhn.ca (S.V.B.); 4Department of Medical Biophysics, University of Toronto, Toronto, ON M5G 1L7, Canada; 5Department of Radiation Oncology, University of Toronto, Toronto, ON M5T 1P5, Canada; 6Department of Pediatrics, University of Calgary, Calgary, AB T2N 4N1, Canada; 7Ohlson Research Initiative, Arnie Charbonneau Cancer Institute, Cumming School of Medicine, University of Calgary, Calgary, AB T2N 4N1, Canada

**Keywords:** OSCC, polo-like-kinase-1, γ-radiation therapy, volasertib

## Abstract

Polo-like-kinase-1 (PLK-1) is a serine/threonine kinase that regulates the cell cycle and acts as an oncogene in multiple cancers, including oral squamous cell carcinoma (OSCC). The loss of PLK-1 can inhibit growth and induce apoptosis, making it an attractive therapeutic target in OSCC. We evaluated the efficacy of PLK-1 inhibitors as novel, targeted therapeutics in OSCC. PLK-1 inhibition using BI6727 (volasertib) was found to affect cell death at low nanomolar concentrations in most tested OSCC cell lines, but not in normal oral keratinocytes. In cell lines resistant to volasertib alone, pre-treatment with radiotherapy followed by volasertib reduced cell viability and induced apoptosis. The combinatorial efficacy of volasertib and radiotherapy was replicated in xenograft mouse models. These findings highlight the potential of adding PLK-1 inhibitors to adjuvant therapy regimens in OSCC.

## 1. Introduction

Oral squamous cell carcinoma (OSCC) is the most common head and neck neoplasm, afflicting almost 300,000 individuals annually around the globe [1]. Despite significant advances in conventional therapeutic regimens, the 5-year survival for OSCC patients remains poor at around 50%, with a progressive increase in the mortality rate with advancing stage at diagnosis [2,3]. This can be attributed to several factors, including late detection, a lack of effective methods for prognostication, the presence of lymph node metastasis, loco-regional recurrence and a lack of effective targeted therapeutics [2,4]. OSCC is treated with surgery followed by adjuvant radiotherapy and/or chemotherapy. However, radio-resistance is common in OSCC patients [5]. In recent years, immune checkpoint blockade and other immunotherapy regimens have gained popularity in the treatment of recurrent/metastatic OSCC, but they are effective in only a minority of patients [6]. Therefore, new and more effective treatment strategies with long-term remission potential are required [7].

Polo-like-kinases (PLKs) are a group of mitotic kinases involved in the regulation of the cell cycle [8]. PLK-1, the most studied member of the PLK family, is a serine/threonine kinase that regulates transition into mitosis and maintains the G2/M checkpoint. Additionally, PLK-1 synchronizes the cycles of the centrosome and the cell, and it plays a crucial role in assembling the spindle and segregating chromosomes. It also performs several functions at the spindle midzone and in the process of abscission. Beyond these roles, PLK-1 aids in DNA replication and is actively involved in cytokinesis and meiosis. The fundamental importance of PLK-1 lies in its ability to precisely control cell division, thereby ensuring genomic stability during mitosis, spindle formation, and the DNA damage response [9]. Consequently, PLK-1 acts as an oncogene in multiple cancers, with its overexpression being correlated with poor prognosis and its knockdown leading to cancer cell death [10,11]. In head and neck cancer, PLK-1 levels positively correlate with lymph node metastasis and poor prognosis, thus implicating PLK-1 as a prognostic marker [12,13,14]. Radio-resistant OSCC cells show aberrant antioxidant stress regulation and the suppression of the apoptotic and DNA damage response pathways [5,15,16]. PLK-1 inhibition has been shown to regulate similar processes, thus raising the possibility that PLK-1 inhibition may mitigate radio-resistance in OSCC [10,12,17,18,19].

BI2536 is an early PLK-1 inhibitor that functions by binding competitively to the N-terminal ATP-binding domain, leading to defective mitotic spindle formation, G2/M arrest, and subsequent apoptosis [20]. However, despite preliminary success in preclinical experiments [20], BI2536 has failed to show efficacy in clinical trials due to a poor terminal half-life in solid tumors (~50 h) and minimal penetration into the microenvironment of solid tumors [21,22,23]. A second-generation PLK-1 inhibitor (BI6727/Volasertib^®^) showed greater penetration into the solid tumor microenvironment and increased specificity for PLK-1 [21,24,25]. Similar to BI2536, Volasertib blocks the proliferation of cancer cells by inducing G2/M arrest through defective mitotic spindle formation followed by apoptosis [24].

In glioblastoma models, the therapeutic efficacy of radiation synergistically increases when volasertib is added [26,27]. In head and neck cancer, the effects of PLK-1 mRNA depletion is enhanced with the addition of radiotherapy, indicating that depleting PLK-1 may enhance the cytotoxic effects of radiotherapy [18]. Therefore, the combinatorial therapy of volasertib with ionizing radiation (IR) could be a possible therapeutic strategy able to augment the effect of volasertib *in vitro* and *in vivo*. In this study, we provide evidence for the use of volasertib as a targeted therapeutic in OSCC. Further, our results support the addition of volasertib to radiotherapy regimens that are routinely used in OSCC patients.

## 2. Materials and Methods

### 2.1. Cell Culture

Skin fibroblast cell strain Hs68, oral keratinocyte strain OKF6/TERT, and the OSCC cell lines CAL-27 and CAL-33 were obtained from American Type Cell Culture (ATCC; Manassas, VA, USA). The OSCC cell lines UMSCC1, UMSCC2, UMSCC7, UMSCC29, UMSCC43, UMSCC57, UMSCC59, and UMSCC103 were obtained from the University of Michigan. All cell lines were grown in high-glucose Dulbecco’s modified eagle medium (DMEM) (Gibco, Waltham, MA, USA) supplemented with 10% Fetal Bovine Serum (FBS; Invitrogen, Waltham, MA, USA), 1% penicillin–streptomycin (Invitrogen) and 1X Non-Essential Amino Acid Solution (MilliporeSigma, Burlington, MA, USA) in a humidified environment at 37 °C and 5% CO_2_. Every 2–3 days, the cells were sub-cultured (sub-cultivation ratio 1:5) using trypsinization (Trypsin, Invitrogen). All cells were used within 25 passages and regularly tested for mycoplasma contamination using the Mycoalert kit (Lonza, Basel, Switzerland).

### 2.2. Drug Screen

In total, 146 drugs (Appendix A) were used to screen the CAL-27 cell line. Three PLK-1 inhibitors were included in this screen. Multiple different concentrations of each drug from 1 nM to 10 μM were used, and the Alamar blue assay (Invitrogen) was employed to assess cell viability after 72 h. The IC_50_ value for each drug was calculated by plotting the dose–response curve using GraphPad PRISM software (ver. 8.0.1)and calculating the concentration of the drug that was required to achieve 50% cell toxicity.

### 2.3. The Cancer Genome Atlas Analysis

The gene expression data were downloaded from The Cancer Genome Atlas (TCGA; https://gdac.broadinstitute.org/; accessed on 2 June 2021). The clinical outcomes data were downloaded from Sage Synapse (https://www.synapse.org/; accessed on 2 June 2021). Boxplots were used to compare the mRNA levels of PLK-1, PLK-2, PLK-3, PLK-4 and PLK-5 between the OSCC and normal oral cavity squamous epithelium. Statistical significance was tested using the Wilcoxon’s rank-sum test. Kaplan–Meier survival curves were used to study the association between overall survival (OS) and the mRNA levels of PLK family members. The log-rank test was used to test for statistical significance.

### 2.4. Cell Viability Assay

OSCC cells were passaged as stated above and plated at a density of 1.0 × 10^4^ per well in a 96-well plate. After 24 h, the drugs (BI2536 and BI6727) diluted in DMSO were added at various concentrations (1 pM–10 μM); DMSO alone was used as a negative control. Plates were incubated for 72 h, and the cell viability was measured using the Alamar blue assay. The cell survival curves were plotted and the IC_50_ was calculated using GraphPad Prism software. Three technical replicates were used for each drug concentration. All experiments were repeated at least three times (biological replicates).

### 2.5. Microscopy

1 × 10^4^ OSCC cells were seeded onto 8-well glass chamber slides in triplicate. After 24 h, volasertib was added, starting from concentrations of 10 µM to 1 µM. Bright field images were collected after 72 h of treatment using Olympus CKX53 coupled to a EP50 camera. Fluorescence images were collected using a Zeiss Axiovert 200 microscope at the Charbonneau Cancer Institute Microscopy Facility.

### 2.6. Radiation Assay

OSCC cells were plated onto 6-well tissue culture plates (5.0 × 10^4^ cell/well) and irradiated after 24 h with various IR doses from 0 to 10 Gy using a Gamma cell 1000 Tissue Irradiator (MDS Nordion; Ottawa, ON, Canada) with a Caesium-137 source. This experiment was performed to understand the efficacy of radiation alone on OSCC cell lines. These cells were then tested for cell viability using the Alamar blue assay at 24, 48, 72 h.

### 2.7. Cell Cycle Analysis

An equal number of OSCC cells (2.0 × 10^5^) were plated in each well of 6-well plates, and volasertib was added 24 h after plating. Cells were then harvested at 24, 48 and 72 h by trypsinization and washed with 1× PBS. Cells were then fixed with 50% ethanol overnight at 4 °C, and then the stages of the cell cycle were analyzed using Fx violet stain (Invitrogen) via flow cytometry using a FACSAria instrument. The cell cycle was then analyzed using the associated software (FACSDiva v 6.1.3). Cells were assigned to different phases of the cell cycle, i.e., G1, S, G2/M or hyperploid based on their nuclear DNA content.

### 2.8. Combinatorial Therapy with Volasertib and Radiation

Resistant cell lines that did not reach an IC_50_ for volasertib (UMSCC7 and UMSCC43) were used for combination studies. 5.0 × 10^4^ cells were seeded in 60 mm plates. Two methods were used after 24 hrs of seeding: (1) In Section 1, the cells were subjected to radiations (2 Gy and 4 Gy) first and volasertib (50 nM) was administered 24 h after radiation treatment. (2) In Section 2, the cells were subjected to volasertib first, followed by radiation after 24 h of volasertib. The drug concentration and radiation dose remained the same as that in Section 1. The cell viability in both cases was assessed 24, 48 and 72 h after final treatment using the Alamar blue assay as described above.

### 2.9. Immunoblot Analysis

The cells were lysed and sonicated; protein was extracted using NETN buffer (150 mM NaCl, 1 mM EDTA, 50 mM Tris-HCl pH 7.5, 1% (*v*/*v*) NP40), and protease inhibitor cocktail (1× PIC; MilliporeSigma) was added to the protein lysates. The cells were suspended in 200 µL of buffer and the cells were sonicated 2–3 times for 5 s. The amount of proteins was estimated using the Bradford assay. An equal amount of protein was then loaded into each well of the appropriate percentage (Tris-Glycine) protein gel and then transferred onto a Polyvinylidene difluoride (PVDF) membrane. The membrane was blocked for one hour at room temperature with 5% skim milk powder in PBS containing 0.1% Tween-20 (MilliporeSigma). The blots were incubated with primary antibodies against PLK-1 (#4513), pPLK-1 (#5472), PARP-1 (#9542), and cleaved caspase-3 (#9661) (Cell Signaling Technology, Danvers, MA, USA) overnight at 4 °C, washed and probed with the appropriate secondary antibodies conjugated to horseradish peroxidase (HRP) (Abcam, Cambridge, UK); this was followed by visualization through ECL (VWR, Radnor, PA, USA). β-actin (#C4; Santa Cruz Biotechnology, Dallas, TX, USA) was used as a loading control.

### 2.10. In Vivo Xenograft Assay

NRG mice were injected subcutaneously on the left flank with 5.0 × 10^5^ dissociated OSCC patient-derived xenograft (PDX; #68614 [28]) tumor cells and monitored biweekly for tumor growth. Once tumors grew to 50–100 mm^3^ in size, the mice were separated into 4 treatment cohorts: (1) vehicle only (n = 3), (2) ionizing radiation (IR) + vehicle (n = 7), (3) drug alone (n = 5), and (4) IR + drug (n = 7). Using a 320 kV small animal irradiator (X-RAD 320, Precision X-Ray), 5 Gy of IR was delivered on 5 consecutive days, for a total dose of 25 Gy being delivered to the tumor. Volasertib 10mg/kg in vehicle (25% DMSO, 25% ethanol, 50% saline) was delivered via tail vein injection on days 1, 4, 6, 10, and 14. Tumor measurements were recorded biweekly, and their volume was calculated using the ellipsoid formula. The primary endpoint was regrowth to 5× initial tumor volume. The time to endpoint was analyzed using the Kaplan–Meier method, and the groups were compared using log-rank tests. The nadir was defined as the smallest tumor volume after the initiation of treatment, and the groups were compared using Mann–Whitney U tests.

### 2.11. Statistical Analysis

A minimum of three biological replicates were performed for each experiment, except where stated. Student’s *t*-test (for two groups) or a one-way analysis of variance (ANOVA) followed by the Student–Newman–Keul’s post-test (for more than two groups) using GraphPad Prism software were performed to evaluate the statistical significance of the data. *p*-values ≤ 0.05 were considered statistically significant.

## 3. Results

### 3.1. High-Throughput Viability Assay Identifies PLK-1 Inhibitors as Effective Drugs in the CAL-27 OSCC Cell Line

In order to identify effective targeted therapies in OSCC, we performed a high-throughput screen in CAL-27 cells with a panel of 146 FDA-approved drugs (Appendix A). The Alamar blue cell viability assay revealed CAL-27 cells to be exquisitely sensitive to PLK-1 inhibitors (BI2536 and volasertib) at an IC_50_ of 0.01 µM (Appendix A). GSK461364, another PLK-1 inhibitor, exhibited a higher IC_50_ of 1 µM (Appendix A). Given the low efficacy of GSK461364, we decided to perform further experiments with BI2536 and Volasertib.

### 3.2. Overexpression of PLK-1 Is Associated with Significantly Worse Survival

In order to further rationalize the use of PLK-1 inhibitors in OSCC, we asked whether PLK genes are overexpressed in OSCC compared to the normal oral cavity squamous epithelium (OCSE). We analyzed the mRNA expression data of PLK family members in OSCC patients from The Cancer Genome Atlas (TCGA). We found that, except for *PLK-5*, all other members of the PLK family are significantly overexpressed in OSCC compared to normal OCSE (Figure 1A). Also, in OSCC patients, a higher expression of *PLK-1*, *4* and *5* is associated with significantly worse overall survival (Figure 1B). Patients with high *PLK-1* expression (>985.4) had a median survival of ~900 days, while patients with below-median PLK-1 expression (<985.4) had a median survival of ~2700 days (log-rank *p*-value = 0.0054). Therefore, we conclude that *PLK-1* is overexpressed in OSCC patients and that its overexpression is associated with worse overall survival.

### 3.3. Potent Cytotoxic Effects of PLK-1 Inhibitors BI2536 and Volasertib on OSCC Cell Lines

To determine whether there is a therapeutic window for BI2536 and volasertib, cytotoxicity assays were performed using 0 to 10 μM drug concentrations on a normal oral keratinocyte immortalized with telomerase (OKF6/TERT2) and a normal skin human diploid fibroblast (Hs68). No cytotoxicity was observed at low concentrations of BI2536 and volasertib. Higher concentrations above 1 μM of BI2536 and volasertib reduced cell viability only by 20 to 30 percent and did not reach IC_50_ in both cell strains (Figure 2A). Identical cytotoxicity assays were performed on the OSCC cell lines (Figure 2B). Of the nine OSCC cell lines tested, seven cell lines (CAL-27, UMSCC51, CAL-33, UMSCC1, UMSCC59, UMSCC29, and UMSCC103) were found to be sensitive to BI2536 and volasertib, based on the IC_50_ values (range 4–200 nM). Notably, four cell lines (CAL-27, UMSCC51, CAL-33, UMSCC1) were particularly sensitive to the PLK-1 inhibitors (IC_50_ from 3–10 nM). The other three sensitive cell lines—UMSCC59, UMSCC29, UMSCC103—also displayed sensitivity but required higher concentrations of the drugs to achieve the same cytotoxic effect, with IC_50_ values between 80 to 200 nM. Finally, two cell lines (UMSCC7 and UMSCC43) displayed almost complete resistance to BI2536 and volasertib, as the IC_50_ could not be reached. The IC_50_ values for BI2536 and volasertib in the different normal and OSCC cell lines tested are detailed in Appendix A.

Given that BI2536 has failed to show efficacy in human trials, we focused further efforts on characterizing the effects of volasertib in OSCC [29]. A comparative morphological analysis was conducted using a volasertib-sensitive (UMSCC1) and a volasertib-resistant (UMSCC7) cell line to evaluate the cellular changes pre and post treatment. Microscopic analysis was performed on UMSCC7 after treatment with 1 nM, 5 nM and 10 nM of volasertib; DMSO alone was used as a control. None of the three concentrations of volasertib induced significant morphological alterations or cell death in the UMSCC7 cells, corroborating their resistance to volasertib (Appendix A). In contrast, a significant cytotoxic effect was observed in UMSCC1 cells, with the majority undergoing cell death. A morphological examination revealed characteristics typical of apoptosis, such as cell rounding and membrane blebbing. This apoptotic response was also pronounced at the IC_50_ value of 5 nM for UMSCC1 cells (Appendix A), underscoring the sensitivity of this cell line to volasertib-induced apoptosis.

### 3.4. Volasertib Caused G2/M Arrest in OSCC Cell Lines Sensitive to the Treatment, Whereas Resistant Cell Lines Evaded This Arrest

Cell cycle dysregulation is a hallmark of cancer, and our findings provide insights into the differential responses of the cell cycle to volasertib treatment. In our experiments, the majority of the cells were in the G1/S phase of the cell cycle before treatment (0 h; Figure 3). The cells were treated with the IC_80_ concentration obtained from the volasertib-sensitive cells for 24, 48, and 72 h. In normal diploid skin fibroblasts (Hs68), volasertib did not induce G2/M arrest, consistent with their observed resistance to PLK-1 inhibition (Figure 3A). In contrast, the volasertib-sensitive OSCC cells (CAL-27 and UMSCC1) exhibited pronounced G2/M arrest at 72 h post treatment, implying effective PLK-1 inhibition (Figure 3B,C). In line with observations in normal skin fibroblasts, the resistant OSCC cell lines (UMSCC7 and UMSCC43) did not maintain G2/M arrest, with no cells in this phase at 72 h after treatment, rendering them resistant to volasertib (Figure 3D,E and Appendix A).

A closer look at the cell cycle dynamics revealed that Hs68 was initially arrested at the G2/M phase at 24 h but then transitioned towards the G1/S phase by 72 h, with a noted increase in hyperploidy (Figure 3A). This escape from G2/M arrest could indicate a cell’s intrinsic ability to overcome PLK-1 inhibition. The UMSCC1 cell line, however, maintained G2/M phase arrest from 24 to 72 h after volasertib treatment, reinforcing its sensitivity to the drug (Figure 3B). CAL-27 cells also displayed G2/M arrest at 24 h, transitioned to hyperploidy at 48 h, and showed a slight shift towards G2/M arrest at 72 h (Figure 3B). The volasertib-resistant OSCC cell lines mirrored the Hs68 dynamics, with UMSCC7 and UMSCC43 cells showing escape from G2/M arrest at 72 h (Figure 3D,E). These cells initially displayed the G2/M phase at 24 and 48 h but transitioned towards G1/S and hypoploid, which underscores a resistance mechanism akin to that observed in Hs68 cells (Figure 3E).

### 3.5. Volasertib Reduced PLK-1 and Phospho-PKL1 (pPLK-1) Levels and Triggered Apoptosis Only in Cell Lines Sensitive to the Treatment

Western blot analysis confirmed that volasertib, administered at an IC80 concentration, significantly reduced the PLK-1 and pPLK-1 levels in UMSCC1 and CAL-27 cells within 72 h (Figure 4A,B). This reduction was accompanied by the activation of apoptotic markers, as indicated by PARP cleavage and cleaved caspase-3. In stark contrast, the resistant lines UMSCC7 and UMSCC43 treated with volasertib at a concentration of 50 nM maintained stable PLK-1 and pPLK-1 levels, with no signs of PARP cleavage or the activation of caspase-3 (Figure 4C,D).

In the volasertib-sensitive cell lines (UMSCC1 and CAL27), the levels of PLK-1 and pPLK-1 remained static during the initial six hours post treatment, followed by a progressive decline at 12, 48 and 72 h (Figure 4A,B). This reduction in kinase levels was paralleled by the onset of apoptotic signaling, as evidenced by the absence of PARP cleavage at baseline and its progressive increase starting from six hours post-volasertib exposure. Additionally, cleaved caspase 3 was detected beginning at 12 h, with escalating levels up to 72 h (Figure 4A,B). Therefore, it can be concluded that apoptosis was triggered after the addition of volasertib. These data collectively signify a volasertib-induced, time-dependent apoptotic cascade in sensitive OSCC cells.

Contrastingly, in the volasertib-resistant cell lines (UMSCC7 and UMSCC43), the level of PLK-1 and pPLK-1 remained unaltered after volasertib treatment, suggesting that the drug was ineffective on these kinase pathways. PARP and caspase 3 cleavage was not detected even at 72 h post treatment (Figure 4C,D), supporting the conclusion that apoptotic processes were not initiated in these cell lines following volasertib treatment.

### 3.6. Pre-Treatment with Radiation Counters Cellular Resistance to Volasertib Monotherapy

In our study, cell lines that exhibited complete resistance to volasertib monotherapy, namely UMSCC7 and UMSCC43, along with UMSCC103, which demonstrated a high IC_50_ value for volasertib (94.366 ± 8.17 nM), were subjected to a combination of IR and volasertib to assess the synergistic potential of these regimens. Combinatorial therapy showed a significant increase in cytotoxicity, suggesting a synergistic interaction between IR and volasertib.

When we compared the efficacy of radiotherapy followed by volasertib administration (adjuvant therapy) with the inverse sequence (neoadjuvant therapy), the adjuvant approach resulted in the superior induction of cell death. Specifically, delivering 4 Gy of IR in conjunction with volasertib was markedly more effective than delivering 2 Gy of IR with volasertib, achieving higher rates of cytotoxicity across all three cell lines at 72 h. Section 1, which involved radiotherapy followed by volasertib treatment, exhibited greater cytotoxicity, reducing cell viability to 10–20% at 72 h post treatment. In contrast, Section 2 showed less pronounced cell death, with 30–50% of the cells remaining viable at the same time point. Figure 5A depicts the Alamar blue cytotoxicity assay for UMSCC-43. The results for UMSSC 7 and UMSCC103 are summarized in Appendix A, respectively. The differential outcomes between the two sets of treatment regimens underscore the importance of treatment sequencing in enhancing the therapeutic efficacy of volasertib.

### 3.7. Combinatorial Therapy with IR Followed by Volasertib Arrests Cells in S-Phase

The resistant cell lines treated with volasertib alone did not display any significant G2/M arrest. Radiation (2 Gy or 4 Gy) alone increased the percentage of cells in the S-phase at 48 h. However, when IR was applied prior to volasertib treatment, a substantial increase in the proportion of cells arrested in the S-phase was observed, particularly at 72 h post treatment. Figure 5B summarizes the cell cycle progression of the volasertib-resistant UMSCC43 cell line from 0 to 72 h and Appendix A provides a similar analysis for UMSCC7. Our results show a notable accumulation of cells in the S-phase starting at 24 h after combination therapy. Interestingly, while treatment with volasertib alone initially caused significant G2/M arrest in UMSCC7 cells, these cells managed to escape this arrest 48 h post treatment (Appendix A). UMSCC43 showed early G2/M arrest that dissipated after 24 h (Figure 5B). Notably, IR alone was capable of inducing S-phase arrest in both the UMSCC7 and UMSCC43 cell lines, although the effect was not as pronounced as when IR was combined with volasertib.

### 3.8. Sequential IR and Volasertib Treatment Achieves Reduced Growth of Xenografted OSCC Tumors

Cells dissociated from an OSCC PDX were injected into the flank of NRG mice; when the tumors reached 0.5–1cm^3^ in size, they were exposed to sequential IR doses for a total of 25 Gy. This was followed by five treatments of volasertib in the drug alone or IR+volasertib groups. Notably, the combined IR and volasertib treatment regimen resulted in a significant (*p* = 0.0035) reduction in the nadir size compared to IR alone (Figure 6A). Furthermore, there was an increase in the time taken to reach the nadir for the IR+volasertib group compared to IR alone, although this did not reach statistical significance (Figure 6B). The average time to endpoint was prolonged to 84 days in the volasertib + IR group compared to 71 days for the vehicle + IR group (log-rank *p* = 0.14). Volasertib alone also appeared to extend the time to endpoint, with 28 days observed compared to 17 days for the vehicle control, although this difference was not statistically significant (log-rank *p* = 0.09) (Figure 6C). There was also a delay in tumor regrowth post IR treatment when volasertib was added (Figure 6D).

## 4. Discussion

In this study, by using an unbiased high-throughput drug screen, we identified that PLK-1 inhibitors, particularly volasertib, show potent cytotoxic effects in OSCC, both *in vitro* and *in vivo*. The results from the *in vitro* studies indicate that most OSCC cell lines were highly sensitive to volasertib. The IC_50_ value of volasertib observed in this study falls in the range seen in other reports evaluating PLK-1 inhibition in head and neck cancer cells [30,31]. Radiation is routinely used in the post-operative setting in OSCC [7]. Therefore, resistant cell lines in which an IC_50_ value could not be reached with volasertib treatment alone were treated with a combination of radiotherapy and volasertib. Combination therapy (radiotherapy followed volasertib) proved to be more lethal to cells *in vitro* than either volasertib or radiation alone. *In vivo*, the combination treatment of xenograft tumors with radiation followed by volasertib led to a reduced tumor size and a delay in tumor volume increase (Figure 7).

The differential cytotoxic response of OSCC cell lines to PLK-1 inhibitors emphasizes the heterogeneity within OSCC and suggests a need for personalized approaches to therapy. The majority of the OSCC cell lines tested were sensitive to volasertib, with notable exceptions displaying almost complete resistance. This sensitivity pattern aligns with the survival data (Figure 1B), further reinforcing the therapeutic potential of PLK-1 inhibition in OSCC.

Volasertib’s ability to induce G2/M arrest in sensitive OSCC cell lines, but not in resistant lines or normal cells, highlights the specificity of its action and suggests a therapeutic window that could spare non-cancerous cells. The lack of significant morphological alterations or apoptosis in resistant lines post-treatment with Volasertib indicates a potential for intrinsic or acquired resistance mechanisms that warrant further investigation. A limitation of this line of investigation is the use of Hs68 fibroblast cells as a surrogate for normal oral cavity cells. However, we obtained similar results in immortalized OKF6/TERT cells derived from normal oral cavity squamous epithelial cells of ectodermal origin.

We found that the phosphorylation of PLK-1 was reduced after volasertib treatment, suggesting that the drug obstructs the kinase activity by competitively targeting the ATP binding pocket. However, resistance to volasertib could arise from multiple mechanisms. Mutations within the ATP binding pocket, particularly in cells with an intact *p53*, could compromise drug efficacy [32,33]. Furthermore, cells lacking *AJUBA* and *RAS* mutations may also exhibit altered drug sensitivity, pointing to the genetic intricacies governing PLK-1 inhibitor resistance [34].

The total endogenous PLK-1 protein levels pre-treatment do not reflect the sensitivity or resistance of the OSCC cell lines used in this study to PLK-1 inhibitors. The results from the western blots indicate that there were negligible differences in the PLK-1 protein levels between the volasertib-resistant and sensitive cell lines. Similarly, no appreciable difference was observed in the pre-treatment pPLK-1 levels between the volasertib-resistant and sensitive cell lines. Given these findings, we explored alternative methods to overcome volasertib resistance. In other cancer cell types, PLK-1 inhibition has been shown to amplify the effects of radiation therapy, which induces double-stranded DNA breaks (DSBs) and activates DNA repair pathways [35,36]. Considering that PLK-1 inhibition is linked to DNA damage signaling [19,37], a promising direction for future research could be to assess the DSB response markers post volasertib treatment, offering potential insights into the overcoming of drug resistance.

However, when the extent of DNA damage surpasses the cellular repair capacity, it may lead to irreversible damage and subsequent apoptosis [26,36]. Our hypothesis was that IR, in concert with volasertib treatment, might effectively push resistant OSCC cells towards apoptotic cell death, thus enhancing the overall cytotoxicity of the treatment regimen [7]. Cells subjected to a combination of IR and volasertib showed decreased PLK-1 and pPLK-1 protein levels. Apoptotic markers also showed activation, thereby indicating the induction of apoptosis. Flowcytometry analysis showed a strong S-phase arrest after combinatorial therapy. A detailed analysis showed that at 48 h following combination therapy, cells underwent irreversible S-phase arrest. IR has been shown to arrest cells in the S-phase before commencing apoptosis [38,39,40]. The inhibition of PLK-1 via volasertib treatment in the S-phase leads to disrupted pre-RC formation and S-phase arrest, leading to caspase activation [17]. Therefore, our combinatorial therapy was more effective when the cells were first shifted towards the S-phase and then treated with volasertib rather than vice versa. Initial administration leads to an unknown resistance to volasertib, which is rendered sensitive when there is a shift of cells towards the S-phase. A further investigation of the role of PLK-1 in the S-phase and DNA damage could highlight the mechanism of this combination therapy.

PLK-1 expression has been positively correlated with aggressive tumor growth and poor prognosis in multiple solid and liquid tumors, including those of the head and neck [11,31]. Consequently, four PLK-1 inhibitors, including volasertib, are now in various stages of clinical trial for multiple neoplasms [31]. The intricate regulation of PLK-1 by various cellular proteins such as p53, cyclins, aurora kinase, and CDKs highlights a complex cell cycle regulatory network that influences PLK-1 activity and inhibition [9].

Furthermore, using a 3D organoid culture model using OSCC cell lines or patient-derived organoids would provide further clues as to the efficacy of volasertib or other PLK-1 kinase inhibitors in the presence of an extracellular matrix and thus dictate the design of more sophisticated *in vivo* xenograft experiments [41,42]. Finally, further investigation is required to further evaluate the efficacy and tolerance of volasertib or other PLK-1 inhibitors in the presence or absence of IR in different *in vivo* OSCC models. These analyses would inform the clinical considerations and potential toxicities associated with the potential translation to the treatment of human patients with PLK-1 inhibitors.

## Figures and Tables

**Figure 1 biomedicines-12-00503-f001:**
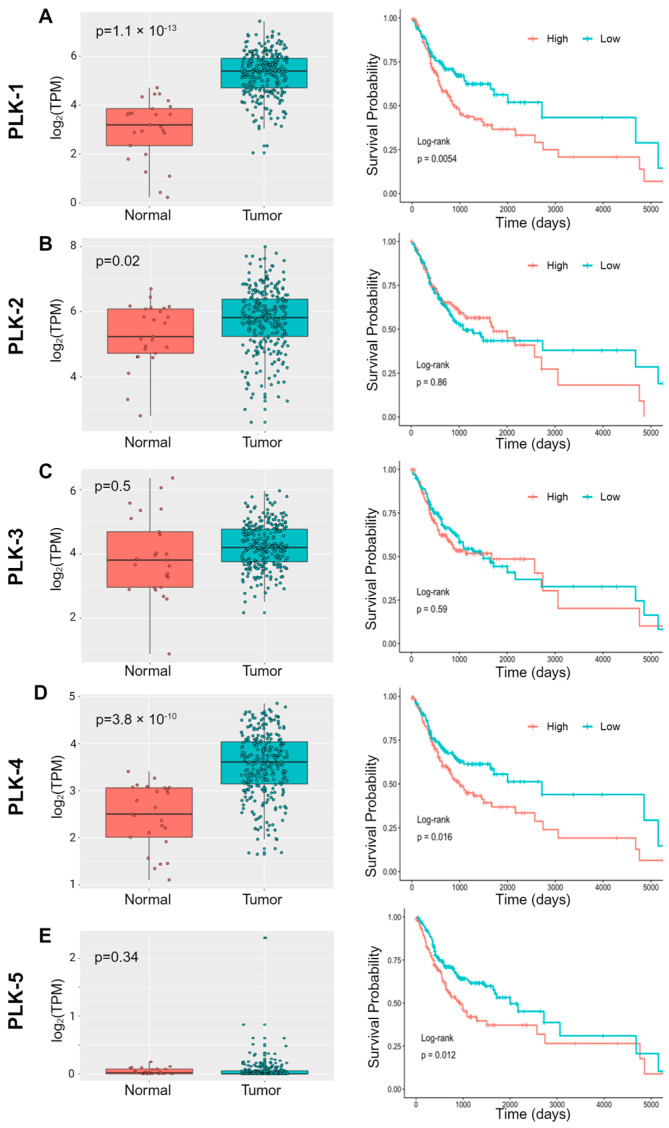
PLK expression in OSCC and correlation to patient survival. (**A**) *PLK-1* is significantly upregulated in OSCC tumor samples when compared to normal oral epithelial tissue, and high *PLK-1* expression is correlated to significantly poor overall OSCC patient survival. (**B**) *PLK-2* is significantly upregulated in OSCC tumor samples when compared to normal oral epithelial tissue; however, *PLK-2* expression is not correlated to overall OSCC patient survival. (**C**) *PLK-3* expression is not significantly altered in OSCC tumor samples when compared to normal oral epithelial tissue, and high *PLK-3* expression is not correlated to overall OSCC patient survival. (**D**) *PLK-4* is significantly upregulated in OSCC tumor samples when compared to normal oral epithelial tissue, and high *PLK-4* expression is correlated to significantly poor overall OSCC patient survival. (**E**) *PLK-5* expression is not significantly altered in OSCC tumor samples when compared to normal oral epithelial tissue; however, high *PLK-5* expression is correlated to significantly poor overall OSCC patient survival.

**Figure 2 biomedicines-12-00503-f002:**
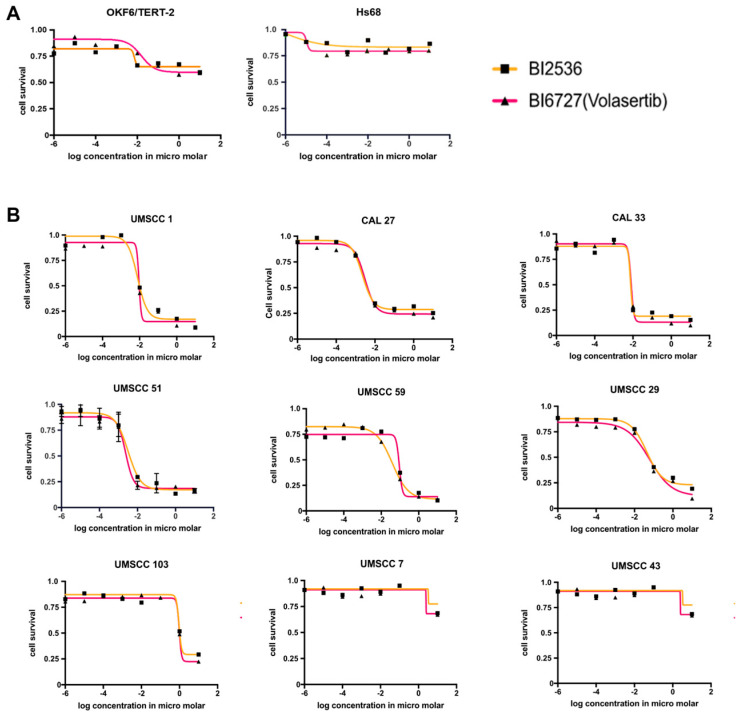
Therapeutic efficacy of PLK-1 inhibitors in human cells. (**A**) The normal oral keratinocyte immortalized with telomerase (OKF6/TERT2) and normal skin human diploid fibroblast (Hs68) cells treated with BI2536 and BI6727 (volasertib) only showed modest cell toxicity. (**B**) The majority of OSCC cells tested showed significant toxicity with PLK-1 inhibition at nanomolar concentrations. However, UMSCC7 and UMSCC43 were almost completely resistant to PLK-1 inhibition.

**Figure 3 biomedicines-12-00503-f003:**
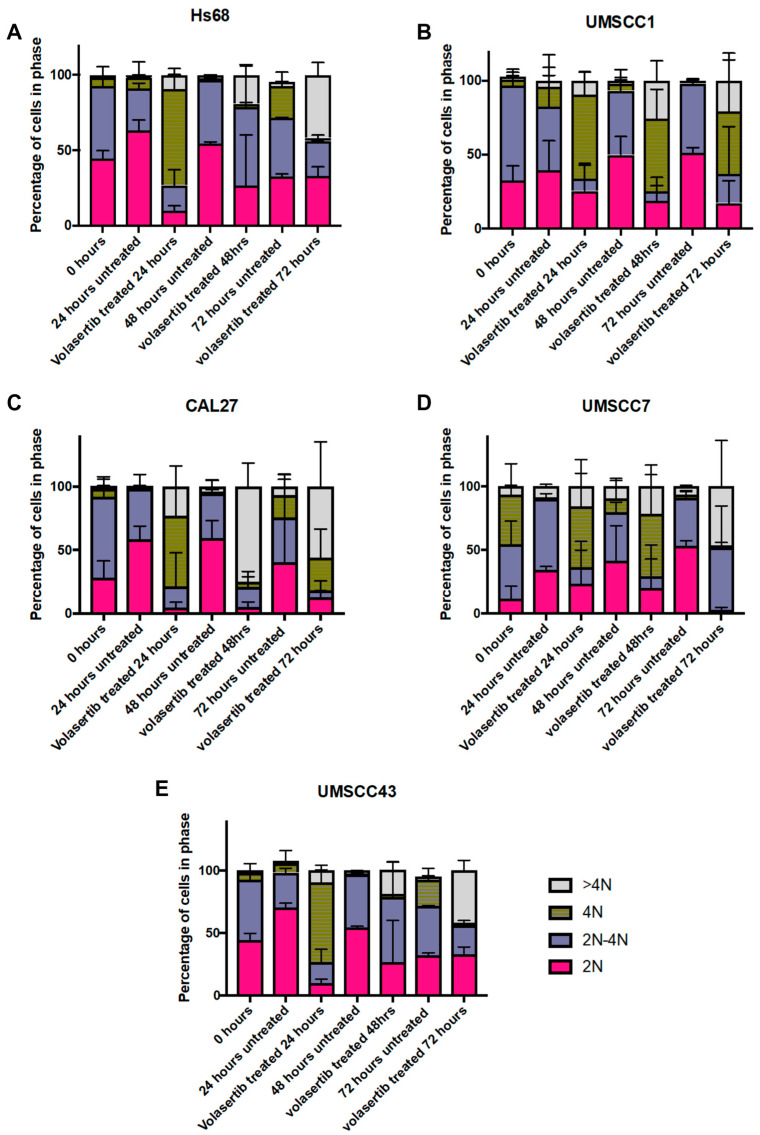
Cell cycle analysis of OSCC cells after volasertib treatment. (**A**) The majority of the Hs68 fibroblasts progressed through the cell cycle even after treatment with volasertib. (**B**,**C**) Volasertib-sensitive CAL27 and UMSCC1 cell lines were arrested at G2/M with treatment. (**D**,**E**) Volasertib-resistant UMSCC7 and UMSCC43 cell lines behaved in a similar way to Hs68 cells, escaping volasertib-induced G2/M arrest at different time points.

**Figure 4 biomedicines-12-00503-f004:**
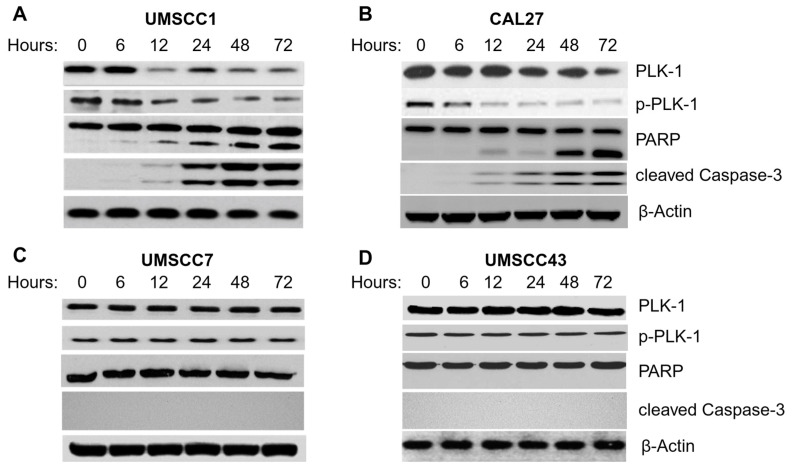
Volasertib induces apoptosis in sensitive cell lines. (**A**,**B**) Lysates from volasertib-sensitive cell lines CAL27 and UMSCC1 showed reduction in p-PLK-1 levels and induction of apoptosis pathways, as evidenced by cleaved PARP and caspase-3 immunoreactive signals. (**C**,**D**) Lysates from the volasertib-resistant cell lines UMSCC7 and UMSCC43 showed maintenance of p-PLK-1 levels and the non-induction of apoptosis pathways, evidenced by the lack of cleaved PARP and caspase-3 immunoreactive signals.

**Figure 5 biomedicines-12-00503-f005:**
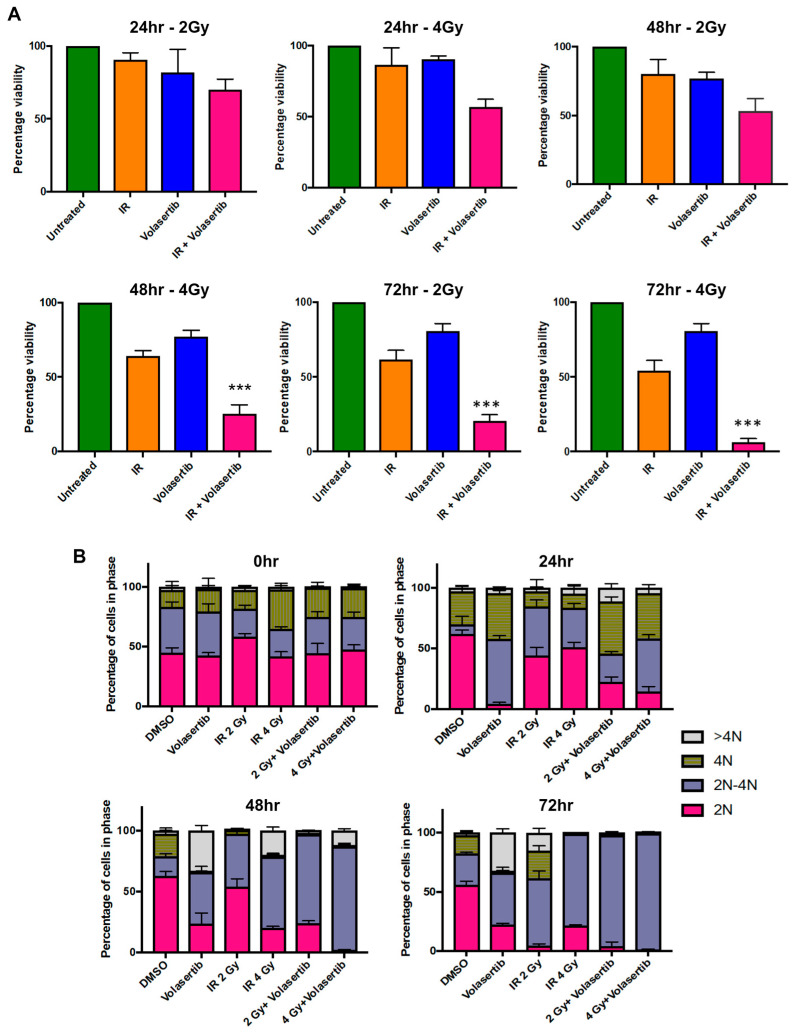
Combination treatment of radiation with adjuvant volasertib. (**A**) In the volasertib-resistant UMSCC43 cell line, ionising radiation (IR) promotes cytotoxicity. After 48 h of volasertib treatment after 4 Gy of IR, there was a significant reduction in cell viability compared to IR alone. After 72 h of combinatorial treatment, a significant reduction in cell viability was observed at both 2 and 4 Gy of IR. (**B**) Cell cycle analysis shows a significant increase in cells in the S-phase at 48 h after IR; this effect was further enhanced in the presence of volasertib. (*** *p* < 0.001).

**Figure 6 biomedicines-12-00503-f006:**
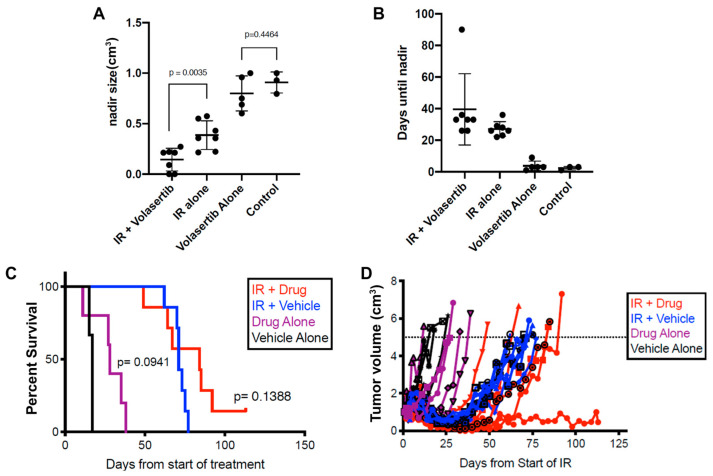
Effect of combination IR and volasertib treatment *in vivo*. (**A**) Adjuvant treatment of volasertib with IR in xenograft mouse tumors led to a significantly smaller tumor size (nadir size) when compared to IR alone. (**B**) The nadir size was reached slower in the IR and IR + volasertib treatment groups compared to the control or volasertib alone groups. (**C**) There was no significant change in survival with or without adjuvant volasertib with IR treatment. (**D**) The tumor volume increase was significantly delayed with IR + volasertib when compared to IR alone.

**Figure 7 biomedicines-12-00503-f007:**
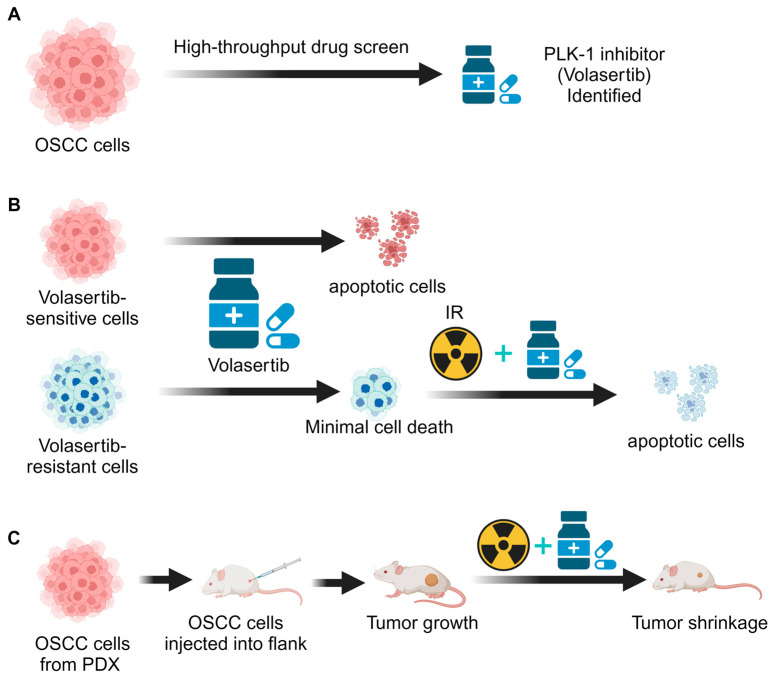
Schematic representation of the study. (**A**) A high-throughput drug screen revealed OSCC cells to be sensitive to several PLK-1 inhibitors, including volasertib. (**B**) OSCC cells can be inherently resistant or sensitive to volasertib. However, exposing the resistant cells with ionization radiation (IR) followed by volasertib leads to significant cell cytotoxicity. (**C**) *In vivo*, OSCC cells injected into a flank of NRG mice form tumors that shrink when treated with IR followed by adjuvant volasertib.

## Data Availability

The original contributions presented in the study are included in the article/Appendix A, further inquiries can be directed to the corresponding author. The raw data supporting the conclusions of this article will be made available by the authors on request.

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
