# Peer review of "Targeting Oral Squamous Cell Carcinoma with Combined Polo-Like-Kinase-1 Inhibitors and γ-Radiation Therapy"

_biomedicines, 2024, doi:10.3390/biomedicines12030503_

Round 1

Reviewer 1 Report

Comments and Suggestions for Authors

Introduction

- Provide more details on the standard treatment regimen for OSCC - typical sequencing of surgery, radiation, chemotherapy. Describe any recent advances.

- Explain the different classifications and clinical stages of OSCC. Mention 5-year survival rates based on stage at diagnosis.

- Give more background on the role of PLK1 in the cell cycle and how its inhibition leads to apoptosis. Include details on the kinase domain structure.

- Elaborate on mechanisms of radioresistance in OSCC. Discuss potential ways PLK1 inhibition could enhance radiation therapy.

Methods

- Provide full details on the origin and culture conditions of the OSCC cell lines used. Include passage numbers.

- Give specifics on the irradiation source, dose rate, and dosing schedule for in vitro radiation experiments.

- Describe the methodology for animal experiments in more depth - source of PDX cells, number of cells injected, flank site, tumor endpoints.

- Provide vendor and catalog information for all antibodies used for Western blotting.

- Include precise details on buffers used for protein extraction and Western blotting protocol.

- Specify the flow cytometry instrumentation used and antibodies utilized for cell cycle analysis.

- Provide details on how IC50 values were calculated from dose-response curves.

- Describe the statistical tests used for comparing data between groups.

The goal should be to include enough methodological detail so the experiments could be reproduced independently by other researchers in the field.

Results:

- In the TCGA analysis, include absolute mRNA expression values for PLK1-5 in OSCC vs normal tissue, not just boxplots. Show statistical significance.  

- For IC50 graphs, add tables with the numeric IC50 values and 95% confidence intervals for each cell line.

- Show Western blots for total PLK1 levels in sensitive vs resistant cell lines.

- For combination therapy, include quantitative graphs for cell viability timecourses, not just 72hr endpoint. Add statistical comparisons.

- Show representative images of tumor histology and immunohistochemistry staining.

Discussion:

- Discuss any correlations noted between PLK1 mRNA levels and response to inhibitors across cell lines.

- Compare IC50 values achieved to other reports on PLK inhibitors in HNSCC models to put efficacy in context.

- Analyze the specific PLK1 mutations or downstream signaling alterations that could cause intrinsic resistance to inhibitors.

- Discuss potential mechanisms of acquired resistance following PLK inhibitor therapy and strategies to overcome it.

- Suggest future in vivo experiments with additional OSCC models and clinical endpoints.

- Propose investigation of biomarker signatures that predict response to combination therapy.

- Discuss clinical considerations and toxicities for translating the combination regimen to patients.

Comments on the Quality of English Language

no

Author Response

We would like to thank the reviewer for their time and comments. Addressing them have improved the quality of the manuscript significantly. Here is a point-by-point response to each comment.

Introduction

1.    Provide more details on the standard treatment regimen for OSCC - typical sequencing of surgery, radiation, chemotherapy. Describe any recent advances.

  • We have provided further details on the standard of therapy and the recent advances in OSCC treatment in the revised manuscript lines 35-39.

  1. Explain the different classifications and clinical stages of OSCC. Mention 5-year survival rates based on stage at diagnosis.

  • The different classification and staging of OSCC are not relevant to our study. Thus, we feel adding this information would be out of the scope of this manuscript.

  1. Give more background on the role of PLK1 in the cell cycle and how its inhibition leads to apoptosis. Include details on the kinase domain structure.

  • We have provided significantly more details about the role of PLK-1 in cell cycle in the revised manuscript lines 43-51. However, we feel that discussing the kinase domain structure of the protein is beyond the scope of this manuscript.

  1. Elaborate on mechanisms of radioresistance in OSCC. Discuss potential ways PLK1 inhibition could enhance radiation therapy.

  • We have provided details about the mechanism of radioresistance development in OSCC and how PLK-1 may play a role in that in the revised manuscript lines 55-58.

Methods

5.    Provide full details on the origin and culture conditions of the OSCC cell lines used. Include passage numbers.

  • We have provided more details about source and culture conditions of all the cell lines used in the study in the revised manuscript subsection “Cell culture” consisting of lines 79-89.

  1. Give specifics on the irradiation source, dose rate, and dosing schedule for in vitro radiation experiments.

  • We believe that we had already provided the requested details in the subsection “Radiation assay”, now in the revised manuscript lines 125-130.

  1. Describe the methodology for animal experiments in more depth - source of PDX cells, number of cells injected, flank site, tumor endpoints.

  • We have updated the subsection “In vivo xenograft assay” subsection in the revised manuscript lines 168-181 to provide the additional information requested.

  1. Provide vendor and catalog information for all antibodies used for Western blotting.

  • We have provided the requested information in the revised manuscript lines 161-166.

  1. Include precise details on buffers used for protein extraction and Western blotting protocol.

  • We believe that we had already provided the requested details in the subsection “Immunoblot analysis”, now in the revised manuscript lines 152-166.

  1. Specify the flow cytometry instrumentation used and antibodies utilized for cell cycle analysis.

  • We have provided the requested information about the instrumentation in the revised manuscript lines 138-139. However, no antibodies were used in this assay and the dye used had already been mentioned in the original manuscript.

  1. Provide details on how IC50 values were calculated from dose-response curves.

  • We have provided further details about the IC50 calculation in the subsection “Drug Screen” in the revised manuscript lines 91-97.

  1. Describe the statistical tests used for comparing data between groups.

  • We have provided a new subsection titled “Statistical Analysis” in the revised manuscript lines 183-188.

Results:

13.  In the TCGA analysis, include absolute mRNA expression values for PLK1-5 in OSCC vs normal tissue, not just boxplots. Show statistical significance.  

  • We believe the boxplots provide the absolute mRNA expression range including standard deviation of the PLK isoforms. The statistical significance (p) values are already provided on the top left of each boxplot.

  1. For IC50 graphs, add tables with the numeric IC50 values and 95% confidence intervals for each cell line.

  • The requested table has already been provided as Table S2 in the original manuscript and we are unsure about what is meant by the 95% confidence interval for such analyses.

  1. Show Western blots for total PLK1 levels in sensitive vs resistant cell lines.

  • We have provided the western blot showing expression of PLK-1 in all relevant cell lines in the original manuscript Figure 4.

  1. For combination therapy, include quantitative graphs for cell viability timecourses, not just 72hr endpoint. Add statistical comparisons.

  • The cell viability after 24, 48 and 72 hours, including relevant statistical significance had already been provided in Figure 5A and Supplementary Figure 4.

  1. Show representative images of tumor histology and immunohistochemistry staining.

  • The in vivo experiment was designed to evaluate the tumor size/volume and mouse viability under various treatment conditions. Tumor histology and immunohistochemistry was beyond the scope of the study and were not performed.

Discussion:

17. Discuss any correlations noted between PLK1 mRNA levels and response to inhibitors across cell lines.

  • The PLK-1 mRNA abundance was not evaluated as part of this study in different cell lines. We only focused on the protein abundance and as mentioned in the manuscript, there were no association observed between protein abundance of PLK-1 and sensitivity to inhibitors.

  1. Compare IC50 values achieved to other reports on PLK inhibitors in HNSCC models to put efficacy in context.

  • We have provided further information of IC50 value of PLK-1 inhibitors in other HNSCC models in the revised manuscript lines 387-388.

  1. Analyze the specific PLK1 mutations or downstream signaling alterations that could cause intrinsic resistance to inhibitors.

  • We had already provided a substantial discussion on the potential mechanisms of PLK-1 inhibitor resistance in the original manuscript, now in lines 416-420 in the revised manuscript.

  1. Discuss potential mechanisms of acquired resistance following PLK inhibitor therapy and strategies to overcome it.

  • We had already provided a substantial discussion on the potential mechanisms of PLK-1 inhibitor resistance in the original manuscript, now in lines 416-420 in the revised manuscript.

  1. Suggest future in vivo experiments with additional OSCC models and clinical endpoints.

  • We have proposed future in vivo experiments to further evaluate the potency and tolerance of PLK-1 inhibitors, particularly volasertib in OSCC, in the revised manuscript lines 459-461.

  1. Propose investigation of biomarker signatures that predict response to combination therapy.

  • We have already proposed this in several sections of the discussion in the original manuscript.

  1. Discuss clinical considerations and toxicities for translating the combination regimen to patients.

  • We have proposed further in-depth in vivo experiments which will potentially highlight the clinical considerations and toxicities of the combination treatment regimen in human patients in lines 459-463 of the revised manuscript.

Reviewer 2 Report

Comments and Suggestions for Authors

Manuscript ID: biomedicines-2802334

Title: Targeting oral squamous cell carcinoma with combined polo-like-kinase-1 inhibitors and γ-radiation therapy

Authors: Subhanwita Sarkar, Rutvij A Khanolkar, Ayan Chanda, Meghan Lambie,
Laurie Ailles, Scott V Bratman, Aru Narendran, Pinaki Bose

.

Reviewer’s comments

This manuscript is well-written regarding anti-tumor effects on the combination and radiotherapy and chemotherapy using a specific PLM1 inhibitor. However, the current format does not reach to the level of acceptance. To enhance the quality of this manuscript, authors need to address the following major and minor points.

Major points:

(1)  Discussion of this manuscript is weak based on the outcomes of this experiment.

Authors should discuss the effect of the PLM1 inhibitor on solid tumors in other tissues/organs. Or, authors should discuss why there have been many reports on head and neck squamous cell carcinoma, compared with other tissues/organs.

(2)  Authors need to discuss how the different experimental design, such as an organoid or a 3D in vitro model, could have a potential impact on the effects of the PLM1 inhibitor because all of the experiments were done in a 2D cell culture in this study.

(3)  As examined for cytotoxic screening in this study, cell cycle analysis of not only Hs68 cells but also OKF6/TERT-2 needs to be done and added in this study.

(4)  The presentations of cell cycle analysis (Fig 3, 5B and S4) is very difficult to read the trends. This reviewer strongly recommends add representative histograms.

(5)  It is not clear how many samples were measured in this study although error bars were shown in many figures.

(6)  The statements on statisitical analysis on cell cycle analysis and viability are missing.

(7)  It would be helpful for readers to add the scheme of conclusion in this study as a new Figure 7.

Minor points

(1)  Authors need to describe the “origin” and “malignant grate” of the oral cancer cells used in this study.

(2)  Authors should use  ×, not X, for number of cells.

(3)  Line 59, and 147: Please use “full name” for abbreviations (IR and PDX) on first appearance in the main text.

(4)  Line 97: Please describe the number of chambers used in this study.

(5)  Line 104 and 122: Please use 5.0 × 104.

(6)  Line 116: Please describe the name of equipment of FCM.

(7)  Line 122: Please use 60 mm, not 6 cm.

(8)  Line 308 and 320: “Cell” should be “cell”.

(9)  Line 313; the result of UMSCC-103 is missing in the Fig S4.

(10)        Line 335 and 340: Please add “Fig”.

(11)        In Figure 3, the data of “0 hours” should be deleted.

(12)        In Figure 6, the location of “B” and “C” should be switched.

(13)        Please add “unit” on Y-axis in Figures 6.

(14)        In Figure 6B, this revierwer does not think the legend is consistent with the figure because it looks like the control group reached the nadir first.

Comments on the Quality of English Language

no

Author Response

We thank the reviewer for the positive remarks about the manuscript. We have provided detailed responses to the points raised in the following section.

 Major points:

(1)  Discussion of this manuscript is weak based on the outcomes of this experiment.

Authors should discuss the effect of the PLM1 inhibitor on solid tumors in other tissues/organs. Or, authors should discuss why there have been many reports on head and neck squamous cell carcinoma, compared with other tissues/organs.

- We have provided discussion about PLK-1 inhibitor usage in other carcinomas, including head and neck in the revised manuscript lines 450-453

(2)  Authors need to discuss how the different experimental design, such as an organoid or a 3D in vitro model, could have a potential impact on the effects of the PLM1 inhibitor because all of the experiments were done in a 2D cell culture in this study.

      - We have provided discussion about the benefits of using 3D organoid models to test PLK-1 inhibitors in OSCC in the revised manuscript lines 456-459. However, in the original manuscript the final confirmatory experiment was performed in a xenograft model and not 2D culture as mentioned by the reviewer.

(3)  As examined for cytotoxic screening in this study, cell cycle analysis of not only Hs68 cells but also OKF6/TERT-2 needs to be done and added in this study.

      - We used the Hs68 cell line as it is a “normal” fibroblast line as opposed to OKF6/TERT-2 line which is immortalized by manipulating TERT which have known to have aberrant effects on the cell cycle. Thus, we believe that using only the Hs68 as a representative “normal” cell line is correct for the cell cycle analysis.

(4)  The presentations of cell cycle analysis (Fig 3, 5B and S4) is very difficult to read the trends. This reviewer strongly recommends add representative histograms.

      - We agree that the bar plots for the cell cycle analysis might be hard to read, however adding 83 histograms would not be suitable for a publication and this was the next best solution. We have however added a new supplementary figure 3 to show a representative histogram that matches the first stacked bar column of Figure 3 (Hs68 cells; 0 hours).

(5)  It is not clear how many samples were measured in this study although error bars were shown in many figures.

      - We have added a statement about sample number in the revised manuscript lines 184-185.

(6)  The statements on statisitical analysis on cell cycle analysis and viability are missing.

      - We have provided a new subsection titled “Statistical Analysis” in the revised manuscript lines 183-188.

(7)  It would be helpful for readers to add the scheme of conclusion in this study as a new Figure 7.

      - We have provided a new figure 7 with a schematic representation of the main findings of this study,

Minor points

(1)  Authors need to describe the “origin” and “malignant grate” of the oral cancer cells used in this study.

(2)  Authors should use  ×, not X, for number of cells.

(3)  Line 59, and 147: Please use “full name” for abbreviations (IR and PDX) on first appearance in the main text.

(4)  Line 97: Please describe the number of chambers used in this study.

(5)  Line 104 and 122: Please use 5.0 × 104.

(6)  Line 116: Please describe the name of equipment of FCM.

(7)  Line 122: Please use 60 mm, not 6 cm.

(8)  Line 308 and 320: “Cell” should be “cell”.

(9)  Line 313; the result of UMSCC-103 is missing in the Fig S4.

(10)        Line 335 and 340: Please add “Fig”.

(11)        In Figure 3, the data of “0 hours” should be deleted.

(12)        In Figure 6, the location of “B” and “C” should be switched.

(13)        Please add “unit” on Y-axis in Figures 6.

(14)        In Figure 6B, this revierwer does not think the legend is consistent with the figure because it looks like the control group reached the nadir first.

            - We have edited the manuscript in the relevant sections as suggested by the reviewer.

Round 2

Reviewer 1 Report

Comments and Suggestions for Authors

all the revisions were addressed. Bests

Author Response

Thank you for your previous comments and time. The manuscript is indeed better by addressing them.

Reviewer 2 Report

Comments and Suggestions for Authors

Reviewer’s comments

Authors well-addressed this reviewer’s comments. Thereforem the quality of this manuscript improved and got closer to being accecpted. Please address the following issues to reach the level of acceptance.

  • In the paragraph added and stated on a 3D organoid, references are needed. By the way, authors should recognize that a xenograft model is completely different from an in vitro 3D culture model as a tool of cancer study.
  • Because Hs68 cells are mesenchymal origin, authors should use epithelial cells (ectodermal origin) as a normal cell control for cell cycle analysis. Instead of adding another experiment, author should critique and discuss this weak experimental design in the section of discussion.
  • In Figure 1, PLK4 and PLK5 are missing on the left.
  • In Figure 3, the bar of “0 hours” (the left end) should be removed from the graph because there are no statements of the data in the main text and legends.
  • As a SFig 3, adding a representative histogram is fine. However, it should be replaced with the one having the range of “>4N”.
  • For the description of cancer cells, authors need to add the “origin of the tissue” such as tongue, oral floor or metastatic lymph nodes and “differentiation grade” such as poorly-dfferentiated carcinoma or highly-differentiated carcinoma.
  • Why are the numbers of UMSCC cells not in order? Is it inappropriate to describe 1, 2, 7, 29, 43, 57, 59, 103?
  • Please add the information of the DMEM glucose level. Is it high or low?
  • Please describe the number of chambers used in this study.
  • Microscopy: 1 × 104
  • How many chambers on one glass slide used? 2 or 4 or 8?
  • “6 well dish” Is this incorrect?  Is a 6-well plate correct?
  • Cell cycle analysis: “volasertive sensitive cells” and “volasertive resistant cells” appear for the first time. Although authors stated what OSCCs are either of them later on, authors need to state the definition of sensitive cells and resistant cells at the first appearance.
  • In vivo xenograft assay: What is the difference between oral cavity squamous cell carcinoma and OSCC? This nomenclature is not clear.

Author Response

Reviewer’s comments:

Authors well-addressed this reviewer’s comments. Thereforem the quality of this manuscript improved and got closer to being accecpted. Please address the following issues to reach the level of acceptance.

  • We thank the reviewer for their comments. Addressing them have improved the quality of the manuscript significantly. Here is a point-by-point response to each comment from the reviewer.

  • In the paragraph added and stated on a 3D organoid, references are needed. By the way, authors should recognize that a xenograft model is completely different from an in vitro 3D culture model as a tool of cancer study.
  • We have added new references 41 and 42 to address this concern. As for the 3D vs xenograft comment, we were merely responding to the reviewer originally suggesting that all our experiments were in 2D as opposed to 3D which we felt was not fully correct. We completely agree that 3D organoid culture and xenograft assays are completely different models of enquiry.
  • Because Hs68 cells are mesenchymal origin, authors should use epithelial cells (ectodermal origin) as a normal cell control for cell cycle analysis. Instead of adding another experiment, author should critique and discuss this weak experimental design in the section of discussion.
  • We have added a critique of using Hs68 cells in our cell cycle analysis in the revised manuscript lines 413-417.
  • In Figure 1, PLK4 and PLK5 are missing on the left.
  • We have edited the figure to add the missing legends.
  • In Figure 3, the bar of “0 hours” (the left end) should be removed from the graph because there are no statements of the data in the main text and legends.
  • We have added a line to discuss the 0 hour bar in the revised manuscript lines 262-264.
  • As a SFig 3, adding a representative histogram is fine. However, it should be replaced with the one having the range of “>4N”.
  • We have edited the figure to show the >4N range. As seen in the stacked bar, there is ~1.5% cells that are in that range.
  • For the description of cancer cells, authors need to add the “origin of the tissue” such as tongue, oral floor or metastatic lymph nodes and “differentiation grade” such as poorly-dfferentiated carcinoma or highly-differentiated carcinoma.
  • We have added the type of cells (fibroblast, oral keratinocyte, or OSCC) in the methods. However, we feel any further description of the cell lines is beyond the scope of this study.
  • Why are the numbers of UMSCC cells not in order? Is it inappropriate to describe 1, 2, 7, 29, 43, 57, 59, 103?
  • We have changed the order as suggested.
  • Please add the information of the DMEM glucose level. Is it high or low?
  • We have added the information in the revised manuscript line 84.
  • Please describe the number of chambers used in this study.
  • We have addressed this and comment 11 in the revised manuscript line 120.
  • Microscopy: 1 × 104
  • We have edited this.
  • How many chambers on one glass slide used? 2 or 4 or 8?
  • “6 well dish” Is this incorrect?  Is a 6-well plate correct?
  • Both the terms are interchangeably used, however we have changed dish to plate in requisite places.
  • Cell cycle analysis: “volasertive sensitive cells” and “volasertive resistant cells” appear for the first time. Although authors stated what OSCCs are either of them later on, authors need to state the definition of sensitive cells and resistant cells at the first appearance.
  • We have removed mention of volasertib sensitivity from the methods section and have described the drug dosing to line 264-265 of the revised manuscript.
  • In vivo xenograft assay: What is the difference between oral cavity squamous cell carcinoma and OSCC? This nomenclature is not clear.
  • We have edited the section to just mention OSCC for clarity.

Round 3

Reviewer 2 Report

Comments and Suggestions for Authors

Authours replied "We have edited the figure to show the >4N range. As seen in the stacked bar, there is ~1.5% cells that are in that range."

However, the population of ">4N range" is not clear and this reviewer does not see any differecens compared with the R1 revision.  Authors need to replace the histogram of cancer cells such as "UMSCC7, 48 hours untreated" with the current of of Hs68. Each range of cell cycle population should be clearly shown in the histogram of "UMSCC7, 48 hours untreated".

Author Response

We have edited the supplementary figure S3 to show a representative histogram for UMSCC7 cells at 0 hours where there is an appreciable number of cells with >4N.